# Multi-Enzymatic Cascades in the Synthesis of Modified Nucleosides: Comparison of the Thermophilic and Mesophilic Pathways

**DOI:** 10.3390/biom11040586

**Published:** 2021-04-16

**Authors:** Ilja V. Fateev, Maria A. Kostromina, Yuliya A. Abramchik, Barbara Z. Eletskaya, Olga O. Mikheeva, Dmitry D. Lukoshin, Evgeniy A. Zayats, Maria Ya. Berzina, Elena V. Dorofeeva, Alexander S. Paramonov, Alexey L. Kayushin, Irina D. Konstantinova, Roman S. Esipov

**Affiliations:** Shemyakin and Ovchinnikov Institute of Bioorganic Chemistry RAS, Miklukho-Maklaya 16/10, 117997 GSP, B-437 Moscow, Russia; ifateev@gmail.com (I.V.F.); kostromasha@mail.ru (M.A.K.); ugama@yandex.ru (Y.A.A.); fraubarusya@gmail.com (B.Z.E.); olga.mikheeva.92@mail.ru (O.O.M.); ldd-94@yandex.ru (D.D.L.); eaz96post@gmail.com (E.A.Z.); berzina_maria@mail.ru (M.Y.B.); iegol2013@gmail.com (E.V.D.); a.s.paramonov@gmail.com (A.S.P.); kayushin.alexej@yandex.ru (A.L.K.); esipov@ibch.ru (R.S.E.)

**Keywords:** purine nucleoside phosphorylase, biocatalysis, enzyme cascade, expression of recombinant enzymes

## Abstract

A comparative study of the possibilities of using ribokinase → phosphopentomutase → nucleoside phosphorylase cascades in the synthesis of modified nucleosides was carried out. Recombinant phosphopentomutase from *Thermus thermophilus* HB27 was obtained for the first time: a strain producing a soluble form of the enzyme was created, and a method for its isolation and chromatographic purification was developed. It was shown that cascade syntheses of modified nucleosides can be carried out both by the mesophilic and thermophilic routes from D-pentoses: ribose, 2-deoxyribose, arabinose, xylose, and 2-deoxy-2-fluoroarabinose. The efficiency of 2-chloradenine nucleoside synthesis decreases in the following order: Rib (92), dRib (74), Ara (66), F-Ara (8), and Xyl (2%) in 30 min for mesophilic enzymes. For thermophilic enzymes: Rib (76), dRib (62), Ara (32), F-Ara (<1), and Xyl (2%) in 30 min. Upon incubation of the reaction mixtures for a day, the amounts of 2-chloroadenine riboside (thermophilic cascade), 2-deoxyribosides (both cascades), and arabinoside (mesophilic cascade) decreased roughly by half. The conversion of the base to 2-fluoroarabinosides and xylosides continued to increase in both cases and reached 20-40%. Four nucleosides were quantitatively produced by a cascade of enzymes from D-ribose and D-arabinose. The ribosides of 8-azaguanine (thermophilic cascade) and allopurinol (mesophilic cascade) were synthesized. For the first time, D-arabinosides of 2-chloro-6-methoxypurine and 2-fluoro-6-methoxypurine were synthesized using the mesophilic cascade. Despite the relatively small difference in temperatures when performing the cascade reactions (50 and 80 °C), the rate of product formation in the reactions with *Escherichia coli* enzymes was significantly higher. *E. coli* enzymes also provided a higher content of the target products in the reaction mixture. Therefore, they are more appropriate for use in the polyenzymatic synthesis of modified nucleosides.

## 1. Introduction

Nucleoside analogs can be synthesized by chemical or enzymatic methods or by a combination of these methods [1,2]. Chemical synthesis is a long multi-stage process involving the introduction and removal of various protective groups in the carbohydrate residue and heterocyclic base, which leads to a significant decrease in the efficiency of the process. Despite the use of selective glycosylation methods, racemic mixtures are formed in the synthesis of nucleosides, which complicates the isolation of target compounds. Enzymatic synthesis has a number of advantages over chemical synthesis: mild reaction conditions, high stereo- and regioselectivity, the minimal use of polluting chemicals and organic solvents, high efficiency, and the absence of undesirable impurities [1,3,4].

Biocatalytic reactions can involve either one enzyme that catalyzes one specific reaction or several enzymes that act sequentially in a cascade of reactions [3]. In recent years, the terms “cascade reactions” or “tandem reactions” have been used to denote poly-enzymatic reactions [5].

Bacterial glycosyltransferases that catalyze the transfer of the pentofuranosyl group to purine or pyrimidine bases are successfully used in the synthesis of various natural nucleoside analogs of biological and pharmaceutical significance. The donor of the pentofuranosyl residue can be a natural nucleoside or its derivative, while natural or modified heterocyclic bases serve as acceptors [3]. Nucleoside phosphorylases (NP) include purine nucleoside phosphorylases (PNP) and pyrimidine nucleoside phosphorylases (PyNP): uridine phosphorylases (UP) and thymidine phosphorylases (TP). One of the most studied polyenzymatic cascades in the synthesis of modified nucleosides is the sequential use of two NPs. The combination of PNP and PyNP (UP or TP) is often used in the enzymatic synthesis of nucleoside analogs. Such a combination allows obtaining purine nucleosides from pyrimidine ones and vice versa [6]. In addition, some enzymes are involved in the biosynthesis of uridyl peptide antibiotics, such as nikkomycin or tunicamycin [7]. One of the enzymes involved in the biosynthesis of the uridyl peptide antibiotic pacidamycin, dehydratase Pac13, can be used as a biocatalyst for the preparation of 3′-deoxynucleosides [8].

The use of nucleoside phosphorylase-producing microorganisms in the synthesis of modified nucleosides of biological and pharmaceutical significance has proven to be highly effective [9,10]. The transglycosylation reaction is usually carried out at 60 °C to inhibit other enzymes—for example, deaminases. At this temperature, some NPs retain most of their activity. However, reactions involving TP are usually carried out at 45 °C, since thymidine phosphorylase loses activity at temperatures above 50 °C.

Temperature restrictions can be overcome by using thermophilic microorganisms such as *Geobacillus stearothermophilus* [11] and *Thermus thermophilus* [4,12,13]. Several strains of *T. thermophilus* that can synthesize purine nucleosides from adenine or hypoxanthine have been selected [12]. An example of pyrimidine nucleoside synthesis is the enzymatic synthesis of 5-methyluridine from inosine and thymine with the participation of immobilized PNP and PyNP *Bacillus stearothermophilus* JTS 859 [14]. A similar approach has been used in the synthesis of several modified nucleosides: cladribine (2-chloro-2′-deoxyadenosine), fludarabine (2′-fluoradenine arabinoside), vidarabine (9-β-D-arabinosyladenine), and others. Partially purified preparations of PNP and PyNP *G. stearothermophilus* B-2194 immobilized on aminopropylated macroporous glass have shown high enzymatic activity and stability at 70 °C and reusability up to 20 times [15]. Immobilized thermophilic nucleoside phosphorylases have been used in the polyenzymatic synthesis of several halogenated nucleoside analogs: cladribine [4], 2-fluoro-2′-deoxyadenosine, and fludarabine [16]. The synthesis of halogenated nucleosides is of particular interest, since they exhibit a broad spectrum of biological activity [9,17].

Recently published data on the activity of the hyperthermophilic PyNP from *Thermus thermophilus* should be noted [13]. This enzyme works in a wide pH range (4-9) at 100 °C and relatively high concentrations of organic solvents (up to 80% *v/v* DMSO and ethylene glycol).

In the case of unnatural nucleoside synthesis, the cascade of ribokinase (RK) → phosphopentomutase (PPM) → *E. coli* nucleoside phosphorylase (NP) appears to be promising [18]. This cascade transforms D-pentose through 5-phosphate into 1-α-phosphate and then into the nucleoside (Scheme 1): 

Phosphopentomutases catalyze the reversible transfer of the phosphate group between the C1 and C5 atoms of ribose or deoxyribose [19]. The resulting α-D-ribose-1-phosphate is a substrate for nucleoside phosphorylases in the synthesis of nucleosides. Tandem enzymatic cascades of nucleoside synthesis using two enzymes (*E. coli* PPM and NP combination) have been adapted for the synthesis of D-ribose, D-arabinose, and 2-deoxy-D-ribose nucleosides [18,20]. The possibility of the polyenzymatic synthesis of 2′-deoxy-β-D-ribofuranosides-8-aza-purine and 8-aza-7-deazapurine from 2-deoxyribose using ribokinase, PPM, and NP has been shown [21].

Cascades of thermophilic microorganism enzymes can also be used for the synthesis of modified nucleotides. Therefore, in 2016, a cascade of thermophilic enzyme *Thermus* species 2.9 was presented for the synthesis of nucleotides from D-pentoses [22].

For the thermophilic cascade synthesis of nucleosides, we previously obtained and described ribokinase from *Thermus* species 2.9 (*T*spRK, gene QT17_05185) [23], purine nucleoside phosphorylases from *Thermus thermophilus* HB27 (*Tth*PNPI, gene TT_RS05405, and *Tth*PNPII, gene TT_RS00985) [24].

The synthesis of natural nucleosides using various nucleoside phosphorylases yields similar results. However, with modified heterocyclic bases or different carbohydrates, the results may vary significantly. It would be promising to obtain recombinant *Tth*PPM and try the complete cascade of thermophilic RK → PPM → PNP to synthesize modified nucleosides. It is essential to carry out a comparative study of the features and efficiency of the cascade syntheses of nucleosides using mesophilic and thermophilic enzymes. In addition, some heterocyclic bases, such as 2-chloroadenine and 2-fluoroadenine, have low water solubility, so the use of thermophilic enzymes might be preferable. Carrying out the reaction with thermophilic enzymes at the operating temperatures of mesophilic enzymes reduces the benefits of this approach; therefore, the reactions with *Thermus thermophilus* enzymes were carried out at a higher temperature.

## 2. Materials and Methods

### 2.1. General Procedures

Tris∙HCl, acetic acid, sodium chloride, glycerol, acrylamide, N,N′-bisacrylamide, ATP disodium salt trihydrate, bromophenol blue, agarose, ethylenediaminetetraacetic acid (EDTA), isopropyl β-D-1-thiogalactopyranoside (IPTG), imidazole, and dimethylformamide (DMF) were purchased from Panreac (Barcelona, Spain). Ethanol was purchased from MedChemProm (Balashikha, Russia). Coomassie Brilliant Blue R-250 was purchased from Bio-Rad. Bacto yeast extract, bacto tryptone, and bacto agar were purchased from Becton Dickinson Biosciences (Franklin Lakes, NJ, USA). NaOH, HCl, pyridine, POCl3, Et2NPh, and hydrogen fluoride pyridine (HF/Py) (70%) were purchased from Merck (Darmstadt, Germany). Sodium persulfate, tetramethylethylenediamine (TEMED), ethidium bromide, and sodium azide were purchased from Helicon (Moscow, Russia). deoxynucleoside triphosphate mix (dNTP) was purchased from Fermentas (Waltham, MA, USA). Dithiothreitol (DTT), phenylmethylsulfonylchloride, magnesium chloride, nickel sulfate, potassium dihydroorthophosphate, 8-azaguanine, allopurinol, 2-chloroadenin, tert-butyl nitrite, and Ni-IDA Sepharose were purchased from Sigma-Aldrich (St. Louis, MO, USA). Sodium dodecylsulfate was purchased from Serva (Heidelberg, Germany). Ampicillin was purchased from AppliChem GmbH (Darmstadt, Germany). Polyethersulfone membranes were purchased from Millipore (Burlington, MA, USA).

Unless otherwise noted, the materials were obtained from commercial suppliers and used without any purification.

9-(2,3,5-Tri-O-acetyl-β-D-ribofuranosyl)-2,6-dichloropurine was synthesized as described in [25,26]. 9-(2,3,5-Tri-O-acetyl-β-D-ribofuranosyl)-2-amino-6-chloropurine was synthesized as described in [27].

NMR spectra were recorded on Bruker Avance II 700 spectrometers (Bruker BioSpin, Rheinstetten, Germany) in DMSO-d6 at 303 K. Chemical shifts in ppm (δ) were measured relative to the residual solvent signals as internal standards (2.508 ppm). Coupling constants (J) were measured in Hz; s—singlet, br.s.—broad signal, d—doublet, m—multiplet, t—triplet. NMR spectra data were provided in the Appendix A.

Liquid chromatography-mass spectrometry was performed on the Agilent 6210 TOF LC/MS system (Agilent Technologies, Santa Clara, CA, USA). 

UV spectra were recorded on a Hitachi U-2900 spectrophotometer (Hitachi, Tokyo, Japan). 

### 2.2. Purification of Recombinant Enzymes

In this study, the following previously obtained recombinant enzymes were used: ribokinase from *Escherichia coli* (*Ec*RK, gene AAA51476) [18,28], phosphopentomutase from *E. coli* (*Ec*PPM, gene AAA97279) [18], purine nucleoside phosphorylase from *E. coli* (*Ec*PNP, gene AAA24401) [29], ribokinase from *Thermus* species 2.9 (*T*spRK, gene QT17_05185) [23], purine nucleoside phosphorylases from *Thermus thermophilus* HB27 (*Tth*PNPI, gene TT_RS05405, and *Tth*PNPII, gene TT_RS00985) [24]. Using the protocols described in their respective publications, batches of purified enzymes were prepared in quantities sufficient for all the experiments.

### 2.3. Cloning, Expression, and Purification of Recombinant TthPPM

The TT_RS08405 gene encoding the phosphopentomutase from *Thermus thermophilus* HB27 was amplified from genomic DNA by PCR with primer PPM-forward (5′-GGTGGTCATATGAAGGCGGTGGCCATCGTTTTG-3′) and PPM-reverse (5′-GGTGGTGCGGCCGCGACGAGGCTCGTTCCGGGG-3′) and cloned into the pET-23a+ expression vector at the NdeI and NotI restriction sites. The resulting plasmid pER-PPM-Tth contained the gene encoding the *Tth*PPM with the C-terminal His-tag. The producing strain *E. coli* NiCo21 (DE3)/pER-PPM-Tth was obtained. Cultivation of the strain was carried out in a lysogeny broth (LB) medium containing 100 µg/mL ampicillin. After reaching an absorbance of A_595_ = 0.8, the cultures were added with IPTG to a final concentration of 0.4 mM, and cultivation was continued for 4 h at 37 °C. After culturing, the cell biomass was separated by centrifugation (2.8 g of wet biomass per liter).

The cell biomass (5.6 g) was disrupted in 50 mM KH_2_PO_4_, pH 6.8, 10 mM EDTA, 1 mM phenylmethylsulphonyl fluoride (PMSF) in a ratio of 1:10 (*w*/*v*) using an ultrasonicator. The clarified cell lysate was heat-treated at 80 °C for 10 min to precipitate the contaminated proteins and DNA. To purify *Tth*PPM, a two-stage technique was developed. In the first step, the protein was purified by anion exchange chromatography using an XK 16/20 column packed with 10 mL of DEAE Sepharose Fast Flow resin and equilibrated with 10 mM KH_2_PO_4_ and 1 mM PMSF, pH 6.8. The target protein was eluted with a linear gradient from 0- to 0.4-M NaCl (100 mL, 2 mL/min). The chromatography eluate was loaded onto an XK 16/20 column packed with 20 mL of Chelating Sepharose Fast Flow and equilibrated with 50 mM Tris∙HCl, pH 8.5. The second chromatography was performed on an XK 16/20 column packed with 20 mL of Chelating Sepharose Fast Flow and equilibrated with 50 mM Tris∙HCl, pH 8.5, and 5 mM EDTA in a linear gradient of 50–200 mM imidazole (100 mL, 2 mL/min). The eluate was concentrated using a 30-kDa cut-off polyethersulfone membrane and then loaded on a HiLoad 16/60 Superdex 75-g size-exclusion column equilibrated with 20 mM Tris∙HCl, pH 8.0, supplemented with 50 mM NaCl, 5% glycerol, and 0.04% (*w*/*v*) NaN_3_. Fractions containing a target enzyme with purity higher than 90% were combined and concentrated to the final concentration of 10.0 ± 0.5 mg/mL.

### 2.4. Analytical Methods

The protein concentration was determined by the Lowry method, using bovine serum albumin (BSA) as the standard [30]. Protein purity was determined using protein electrophoresis in polyacrylamide gel under denaturing conditions [31].

Oligomeric organization of enzymes was assessed by size-exclusion chromatography (SEC) using a Superdex 200 10/300 GL analytical column (GE Healthcare, USA). The analysis parameters were as follows: eluent 25 mM Na_2_HPO_4_, pH 8.0, 150 mM NaCl, and 0.04% (*w*/*v*) NaN_3_; flow rate 0.6 mL/min; time 60 min; detection 220 nm; oven temperature 23 °C; and sample injection volume is 100 µL (50 µg). Gel Filtration Markers MWGF200 and MWGF1000 (Sigma-Aldrich, St. Louis, MO, USA) were used for plotting the calibration curve and calculating the molecular weight.

The analysis of the efficiency of nucleoside synthesis cascades was carried out using the following chromatographic systems:

HPLC system 1: Hitachi Chromaster, column YMC Triart-C18, 50×3.0 mm, 3 µm, eluent 0.1% aqueous trifluoroacetic acid (TFA), detection at 254 nm, and flow rate 0.4 mL/min.

HPLC system 2: Hitachi Chromaster, column Supelcosil LC-18-T, 150×4.6 mm, 5 µm, eluent 0.1M KH_2_PO_4_ pH 6.0, detection at 254 nm, and flow rate 0.4 mL/min.

HPLC system 3: Waters system (Waters 1525, Waters 2489, Breeze 2), column Nova Pak C_18_, 4.6×150 mm, 5 µm, and flow rate 0.5 mL/min. Eluent A: 0.1% TFA/H_2_O and eluent B: 70% MeCN in 0.1% TFA/H_2_O, gradient 0–50% B, 20 min, detection at 254 nm. 

HPLC system 4: Waters system (Waters 1525, Waters 2489, Breeze 2), column Nova Pak C_18_, 4.6×150 mm, 5 µm, and flow rate 1.0 mL/min. Eluent A: 0.1% TFA/H_2_O and eluent B: 70% MeCN in 0.1% TFA/H_2_O, gradient 0–50% B, 20 min, detection at 254 nm. 

HPLC system 5: Waters system (Waters 1525, Waters 2489, Breeze), column Nova Pak C_18_, 4.6×150 mm, 5 µm, and flow rate 1.0 mL/min. Eluent A: 0.1% TFA/H_2_O and eluent B: 70% MeCN in 0.1% TFA/H_2_O, gradient 0–100% B, 20 min, detection at 254 nm. 

### 2.5. Enzyme Assay

*Phosphopentomutase Thermus thermophilus* HB27 *(Tth*PPM*).* Each reaction mixture (0.25 mL, 20 mM Tris∙HCl, pH 8.0) contained 0.5 mM D-ribose 5-phosphate, 0.5 mM adenine, 0.5 mM MnCl_2_, 0.2 µg *Tth*PPM, and 1.4 µg *Tth*PNPII. The reaction mixtures were incubated at 80 °C. Substrate and product quantities were determined using HPLC system 1.

*Phosphopentomutase E. coli (Ec*PPM*).* Each reaction mixture (0.25 mL, 20 mM Tris∙HCl, pH 7.5) contained 0.5 mM D-ribose 5-phosphate, 0.5 mM adenine, 0.1 mM MnCl_2_, 0.26 µg *Ec*PPM, and 1.1 µg *Ec*PNP. The reaction mixtures were incubated at 37 °C. Substrate and product quantities were determined using HPLC system 1.

*Purine nucleoside phosphorylase I Thermus thermophilus* HB27 *(Tth*PNPI*)*. Each reaction mixture (0.25 mL, 25 mM KH_2_PO_4_, pH 8.0) contained 1 mM inosine and 0.025 µg *Tth*PNPI. The reaction mixtures were incubated at 80 °C. Substrate and product quantities were determined using HPLC system 1.

*Purine nucleoside phosphorylase II Thermus thermophilus* HB27 *(Tth*PNPII*).* Each reaction mixture (0.25 mL, 25 mM KH_2_PO_4_, pH 8.0) contained 1 mM adenosine and 0.03 µg *Tth*PNPII. The reaction mixtures were incubated at 80 °C. Substrate and product quantities were determined using HPLC system 1.

*Purine nucleoside phosphorylase E. coli (Ec*PNP*)*. Each reaction mixture (0.25 mL, 25 mM KH_2_PO_4_, pH 7.0) contained 1 mM inosine and 0.11 µg *Ec*PNP. The reaction mixtures were incubated at 37 °C. Substrate and product quantities were determined using HPLC system 1.

*Ribokinase Thermus* species *2.9 (T*spRK*)*. Each reaction mixture (0.25 mL, 20 mM Tris∙HCl, pH 8.0) contained 0.4 mM ATP, 1.2 mM D-ribose, 5 mM MgCl_2_, 50 mM KCl, and 0.3 µg *T*spRK. The reaction mixtures were incubated at 80 °C. Substrate and product quantities were determined using HPLC system 2.

*Ribokinase E. coli (Ec*RK*)*. Each reaction mixture (0.25 mL, 20 mM Tris∙HCl, pH 8.0) contained 0.4 mM ATP, 1.2 mM D-ribose, 5 mM MgCl_2_, 50 mM KCl, and 0.02 µg *Ec*RK. The reaction mixtures were incubated at 37 °C. Substrate and product quantities were determined using HPLC system 2.

### 2.6. Determination of Kinetic Parameters of Phosphopentomutase Thermus Thermophilus HB27 (TthPPM)

Each reaction mixture (100 µL, 20 mM Tris∙HCl, pH 8.0) contained 1 mM adenine, 0.5 mM MnCl_2_, and 0.25 µM α-D-glucose 1,6-bisphosphate, from 0.011 to 1.8 mM *D*-ribose 5-phosphate or 2-deoxy-D-ribose 5-phosphate, 0.02 or 0.05 µg *Tth*PPM, and 0.5 µg *Tth*PNPII. The reaction mixtures were incubated 2 min at 80 °C. HPLC system 1. Each experiment was repeated three times. Kinetic parameters were determined by nonlinear regression analysis using SciDAVis v2.3.0 software. Catalytic constants (k_cat_) were calculated per 1 subunit (43.2 kDa, calculated based on amino acid sequences).

### 2.7. Cascade Reactions

*Testing of D-pentoses.* Each reaction mixture (0.5 mL, 20 mM Tris∙HCl, pH 8.0) contained *D*-pentose (2 mM D-ribose or 2-deoxy-D-ribose or 60 mM D-arabinose or D-xylose or 5 mM 2-deoxy-2-fluoro-D-arabinose); 0.34 mM 2-chloroadenine; 2.5 mM ATP; 2.5 mM MnCl_2_; 50 mM KCl; 2 mM KH_2_PO_4_; and thermophilic or mesophilic recombinant enzymes (ribokinase, phosphopentomutase, and purine nucleoside phosphorylase). The reaction mixtures were incubated at 80 or 50 °C. Substrate and product quantities were determined using HPLC system 1.

*Testing of heterocyclic bases.* Each reaction mixture (0.5 mL, 20 mM Tris∙HCl, pH 8.0) contained 2 mM D-ribose, 0.5 mM heterocyclic base (allopurinol or 8-azaguanine), 2.5 mM ATP, 2.5 mM MnCl_2_, 50 mM KCl, 2 mM KH_2_PO_4_, and thermophilic or mesophilic recombinant enzymes (ribokinase, phosphopentomutase, and purine nucleoside phosphorylase). The reaction mixtures were incubated at 80 or 50 °C. Substrate and product quantities were determined using HPLC system 1.

### 2.8. Nucleosides Synthesis

#### 2.8.1. 2-Chloro-6-methoxy-9-(β-D-ribofuranosyl)purine (**2**)

To a solution of MeONa in MeOH (376 mg, 16.2 mmol Na in 8 mL of MeOH), 1.0 g (2.2 mmol) of dichloride (**1**) was added at room temperature under stirring. In 2 h, the reaction mixture was cooled up to 0 °C, and conc. HCl was added up to pH 7.0. NaCl was filtered off, and the precipitate was washed by hot MeOH (3 × 3 mL). The filtrate was evaporated to a quarter of the volume, 1 mL of H_2_O was added, and the solution was heated to boiling. The clear solution was left at room temperature. The crystals were filtered off and washed with iPrOH and Et_2_O. Yield 560 mg, purity 74%. The desired product was purified by chromatography on silica gel (3 x 30 cm, elution by gradient of MeOH in CHCl_3_ (0–20%), 1 L, with a flow rate of 6 mL/min. Yield 310 mg (0.97 mmol, 44%), purity 92.9% (HPLC system 5).

HRMS (ESI+): *m*/*z* [M + H]^+^ calc. for C_11_H_14_O_5_N_4_Cl 317.0654/319.0624 Cl^35^/Cl^37^, found 317.0636/319.0609, and [Base + H]^+^ calc. for C_6_H_6_N_4_OCl 185.0230/187.0200 Cl^35^/Cl^37^, found 185.0221/187.0189.

^1^H NMR (700 MHz): *δ* = 8.67 (s, 1H, H-8), 5.93 (d, *J*_H1,H2_ = 5.5, 1 H, H-1′), 5.64 and 5.35 (br. s, 2 H, OH-2′, and OH-3′), 5.06 (m, 1 H, OH-5′), 4.52 (t, *J* = 5.1, 1 H, H-2′), 4.16 (t, *J* = 4.2, 1 H, H-3′), 4.12 (s, 3 H, OMe), 3.98 (m, 1 H, H-4′), 3.68 (dd, *J* = 4.0, 12.2, 1 H, H-5′a), 3.58 (dd, *J* = 3.7, 12.0, 1 H, H-5′b). ^13^C NMR (176 MHz): *δ* = 160.78 (C-6), 152.84 (C-4), 151.43 (C-2), 142.72 (C-8), 120.19 (C-5), 87.59 (C-1′), 85.63 (C-4′), 73.85 (C-2′), 70.05 (C-3′), 61.00 (C-5′), 54.89 (OCH_3_). ^15^N NMR (71 MHz): *δ* = 241.3 (N-7), 172.47 (N-9).

#### 2.8.2. 2-Fluoro-6-chloro-9-(2’,3,’5′-tri-O-acetyl-β-D-ribofuranosyl)purine (**5**)

A 70% solution of HF in pyridine (7.5 mL) was cooled to −18 °C, and 3.20 g (7.4 mmol) of monochloride (**4**) was added. Some tert-butyl (2.15 mL) (28.5 mmol) was added dropwise within 8 min, and the reaction mixture was stirred at −18 °C for 30 min and at room temperature for 20 min. The mixture was poured into a glass with ice (150 g) and CaCO_3_ (15 g), filtered off, and the precipitate was washed by CHCl_3_ (150 mL). The organic layer was washed by water (100 mL), 5% NaHCO_3_ (100 mL), water (150 mL), dried by MgSO_4,_ and concentrated. The desired product was isolated by column chromatography on silica gel (2×9 cm; 7 cm of silica gel covered by 2 cm of activated carbon, elution by 200 mL of CHCl_3_, then by 5% MeOH in CHCl_3_, with a flow rate 0.9 mL/min). Yield 1.49 g (3.4 mmol, 46%), purity 88.7% (HPLC system 5).

HRMS (ESI+): *m*/*z* [M + H]^+^ calc. for C_16_H_17_O_7_N_4_FCl 431.0764/433.0735 Cl^35^/Cl^37^, found 431.0766/433.0736.

^1^H NMR (700 MHz): δ = 8.90 (s, 1 H, H-8), 6.31 (d, *J* = 5.2, 1 H, H-1′), 5.92 (t, *J* = 5.5, 1 H, H-2′), 5.61 (t, *J* = 5.6, 1 H, H-3′), 4.44 (m, 1 H, H-4′), 4.40 (dd, *J* = 3.4, 12.2, 1 H, H-5′a), 4.29 (dd, *J* = 5.4, 12.2, 1 H, H-5′b), 2.13 (s, 3 H, Me-Ac3′), 2.06 (s, 3 H, Me-Ac2′), 2.03 (s, 3 H, Me-Ac5′). ^13^C NMR (176 MHz): δ = 169.88 (C=O_Ac5′_), 169.24 (C=O_Ac3′_), 169.09 (C=O_Ac2′_), 156.01 (d, C-2, *J*_C2,F2_ = 215.1), 153.12 (d, C-4, *J*
_C4,F2_ = 16.6), 150.95 (d, C-6, *J*
_C6,F2_ = 17.3), 146.79 (C-8), 130.65 (C-5), 86.09 (C-1′), 79.70 (C-4′), 72.01 (C-2′), 69.63 (C-3′), 62.51 (C-5′), 20.29 (CMe_Ac5′_), 20.22 (CMe_Ac3′_), 20.07 (CMe_Ac2′_). ^15^N NMR (71 MHz): δ = 243.03 (N-7), 169.79 (N-9).

#### 2.8.3. 2-Fluoro-6-methoxy-9-(β-D-ribofuranosyl)purine (**6**)

To a solution of MeONa in MeOH (376 mg, 16.2 mmol Na in 8 mL of MeOH), 1.0 g (2.3 mmol) of nucleoside (**5**) was added at room temperature under stirring. In 2 h, the reaction mixture was cooled up to 0 °C, and conc. HCl was added up to pH 7.0. NaCl was filtered off, and the precipitate was washed by hot MeOH (3 x 3 mL) and concentrated. The desired product was isolated by column chromatography on silica gel (4.6 x 40 cm, elution by a gradient of MeOH in CHCl_3_ (0–30%, 1 L, flow rate 6 mL/min). Yield 358 mg (1.2 mmol, 50%), purity 97.0% (HPLC system 5).

HRMS (ESI+): *m*/*z* [M + H]^+^ calc. for C_11_H_14_O_5_N_4_F 301.0949, found 301.0906, [Base + H]^+^ calc. for C_6_H_6_N_4_OF 169.0525, found 169.0504.

^1^H NMR (700 MHz): *δ =* 8.63 (s, 1 H, H-8), 5.89 (d, *J*_H1,H2_ = 5.5, 1 H, H-1′), 4.52 (t, *J* = 5.3, 1 H, H-2′), 4.16 (t, *J* = 4.3, 1 H, H-3′), 4.13 (s, 3 H, OMe), 3.97 (m, 1 H, H-4′), 3.68 (dd, *J* = 4.1, 12.0, 1 H, H-5′a), 3.57 (dd, *J* = 4.1, 12.0, 1 H, H-5′b). ^13^C NMR (176 MHz): *δ =* 162.16 (d, *J_C6,F_*= 18.0, C-6), 156.97 (d, *J_C2,F_* = 210.5, C-2), 152.89 (d, *J_C4,F_* = 18.8, C-4), 142.53 (d, *J_C8,F_* = 2.0, C-8), 119.35 (d, *J_C5,F_* = 5.0, C-5), 87.57 (C-1′), 85.54 (C-4′), 73.73 (C-2′), 70.01 (C-3′), 60.98 (C-5′), 54.96 (OCH_3_). ^15^N NMR (71 MHz): *δ =* 241.76 (N-7), 172.78 (N-9).

#### 2.8.4. 5-Amino-3-(β-D-ribofuranosyl)triazolo [4,5-d]pyrimidin-7-one (**8**)

8-Azaguanine (28 mg, 0.184 mmol), D-ribose (42.6 mg, 0.284 mmol), MnCl_2_ (44.7 mg, 0.355 mmol), KCl (529 mg, 7.1 mmol), and KH_2_PO_4_ (38.6 mg, 0.284 mmol) were dissolved in Tris∙HCl (20 mM, 142 mL, pH 8.0) under stirring and heating at 80 °C. ATP disodium salt trihydrate (215 mg, 0.355 mmol), *T*spRK (0.6 mg, 11.4 units, concentration in the reaction mixture 4.2 µg/mL), *Tth*PPM (0.2 mg, 2.8 units, concentration in the reaction mixture 1.4 µg/mL), and *Tth*PNPI (0.063 mg, 11.4 units, concentration in the reaction mixture 0.44 µg/mL) were added. The reaction mixture was incubated at 80 °C for 24 h (monitored by HPLC system 1). The solution was neutralized with 2 N HCl and concentrated *in vacuo* to ca. 35 mL. The precipitate was filtered off, and the filtrate was applied to the column (octadecyl–Si 100 polyol (0.03 mm); 4 × 14 cm). The desired product was eluted with 15% ethanol in water. Yield 36.1 mg (0.127 mmol; 69%), purity 98.5% (HPLC system 4).

HRMS (ESI+): *m*/*z* [M + H]^+^ calc. for C_9_H_13_N_6_O_5_ 285.0947; found 285.0930; [Base + H]^+^ calc. for C_4_H_5_N_6_O 153.0525; found 153.0515.

^1^H NMR (700 MHz): *δ =* 6.98 (br. s, 2 H, NH_2_), 5.87 (d, *J* = 4.8, 1 H, H-1′), 5.50 (s, 1 H, OH-2′), 5.15 (br. s, 1 H, OH-3′), 4.78 (m, 1 H, OH-5′), 4.70 (t, *J* = 4.9, 1 H, H-2′), 4.24 (t, *J* = 4.8, 1 H, H-3′), 3.94 (m, 1 H, H-4′), 3.59 (dd, *J* = 4.6, 11.9, 1 H, H-5′a), 3.46 (dd, *J* = 5.9, 11.7, 1 H, H-5′b). ^13^C NMR (176 MHz): *δ =* 151.73 (C-4), 87.75 (C-1′), 85.28 (C-4′), 72.47 (C-2′), 70.30 (C-3′), 61.62 (C-5′). ^15^N NMR (71 MHz): *δ =* 234.90, 156.95.

#### 2.8.5. 1-(β-D-Ribofuranosyl)pyrazolo[3,4-d]pyrimidine-4-one (**9**)

Allopurinol (25 mg, 0.184 mmol), D-ribose (42.6 mg, 0.284 mmol), MnCl_2_ (44.7 mg, 0.355 mmol), KCl (529 mg, 7.1 mmol), and KH_2_PO_4_ (38.6 mg, 0.284 mmol) were dissolved in Tris∙HCl buffer (20 mM, 142 mL, pH 8.0) under stirring and heating at 50 °C. ATP disodium salt trihydrate (215 mg, 0.355 mmol), *Ec*RK (0.018 mg, 5.7 units, concentration in the reaction mixture 0.13 µg/mL), *Ec*PPM (0.57 mg, 2.8 units, concentration in the reaction mixture 4 µg/mL), and *Ec*PNP (0.052 mg, 11.4 units, concentration in the reaction mixture 0.37 µg/mL) were added. The reaction mixture was incubated at 50 °C for 24h (monitored by HPLC system 4). The solution was neutralized with 2-N HCl and concentrated *in vacuo* to ca. 35 mL. The precipitate was filtered off, and the filtrate was applied to the column (octadecyl–Si 100 polyol (0.03 mm); 4 × 14 cm). The desired product was eluted with 10% ethanol in water. Yield 41.6 mg (0.155 mmol; 84%), 98.5% purity (HPLC system 4).

UV (H_2_O, pH 7.0) λmax, nm (ε, M^−1^cm^−1^): 250 (7600), 206 (27600); λmin, nm (ε, M^−1^cm^−1^): 232 (5100).

HRMS (ESI+): *m*/*z* [M + H]^+^ calc. for C_10_H_13_N_4_O_5_ 269.0886; found 269.0849; [Base + H]^+^ calc. for C_5_H_5_N_4_O 137.0463; found 137.0444.

^1^H NMR (700 MHz): δ = 12.25 (s, 1 H, NH), 8.14 (s, 1 H, H-3), 8.12 (s, 1 H, H-6), 6.06 (d, *J* = 4.53, 1 H, H-1′), 4.55 (dt, *J* = 4.57; < 0.5, 1 H, H-2′), 4.21 (t, *J* = 4.8, 1 H, H-3′), 3.91 (m, 1 H, H-4′), 3.58 (dd, *J*_5′a,4_ = 4.57; *J*_5′a,5′b_ = 11.65, 1 H, H-5′a), 3.43 (dd, *J*_5′b,4_ = 5.96; *J*_5′b,5′a_ = 11.78, 1 H, H-5′b). ^13^C NMR (176 MHz): δ = 157.45 (C-4), 153.98 (C-7a), 148.86 (C-6), 135.37 (C-3), 106.23 (C-4a), 88.38 (C-1′), 85.25 (C-4′), 73.31 (C-2′), 70.81 (C-3′), 62.27 (C-5′). ^15^N NMR (71 MHz): δ = 302.20 (N-2), 210.47 (N-7), 204.80 (N-1), 173.67 (N-5).

#### 2.8.6. 9-(β-D-Arabinofuranosyl)-2-chloro-6-methoxypurine (**10**)

Eighty milligrams (0.25 mmol) of riboside (**2**) were dissolved in 5 mL of 10 mM KH_2_PO_4_ (pH 7.0). Fifty microliters of 20 mM Na_2_HAsO_4_ and *Ec*PNP (0.045 mg, 10 units, concentration in the reaction mixture 9 µg/mL) were added. The solution was heated up to 50 °C and kept for 3 h until the complete conversion of riboside (**2**) to base (**3**) (monitored by HPLC System 3, *R*_t_ (**2**) = 7.6 min, *R*_t_ (**3**) = 6.0 min). The reaction mixture was cooled down to 4 °C and kept for 12 h. The precipitate was filtered off, washed with cold (4 °C) 10 mM KH_2_PO_4_ (pH 7.0) (2×5 mL), and suspended in 167 mL of 20 mM Tris∙HCl, pH 8.0.

D-Arabinose (1.5 g, 10 mmol), MnCl_2_ (52.5 mg, 0.417 mmol), KCl (622 mg, 8.35 mmol), and KH_2_PO_4_ (44.7 mg, 0.334 mmol) were added under stirring and heating at 50 °C. ATP (253 mg, 0.417 mmol), *Ec*RK (5.4 mg, 1670 units, concentration in the reaction mixture 32 µg/mL), *Ec*PPM (68.2 mg, 334 units, concentration in the reaction mixture 408 µg/mL), and *Ec*PNP (13.6 mg, 3000 units, concentration in the reaction mixture 82 µg/mL) were added, and the reaction mixture was incubated at 50 °C for 48 h (monitored by HPLC System 3). The solution was neutralized with 2-N hydrochloric acid and concentrated *in vacuo* to ca. 25 mL. The precipitate was filtered off. The filtrate was applied to the column (octadecyl–Si 100 polyol (0.03 mm); 4 × 16 cm), and the product was eluted with a gradient of acetonitrile in water (0–50%, 1 L, flow rate 6 mL/min). Yield 43 mg (0.136 mmol, 54%), white amorphous powder, purity 97.0% (HPLC System 3, *R*_t_ = 9.4 min).

UV (H_2_O, pH 7.0) λmax, nm (ε, M^−1^cm^−1^): 270 (10400).

HRMS (ESI+): *m*/*z* [M + H]^+^ calc. for C_11_H_14_N_4_O_5_Cl 317.0654/319.0624 Cl^35^/Cl^37^, found 317.0636/319.0609; [Base + H]^+^ calc. for C_6_H_6_N_4_OCl 185.0230/187.0200 Cl^35^/Cl^37^, found 185.0221/187.0189.

^1^H NMR (700 MHz): δ = 8.48, (s, 1 H, H-8), 6.27 (d, *J* = 4.8 Hz, 1 H, H-1′), 5.64 (br.s., 0.6 H, 2′-OH), 5.57 (br.s, 1 H, 3′-OH), 5.11 (br.s, 1 H, 5′-OH), 4.24 (m, 1 H, H-2′), 4.14 (m, 1 H, H-3′), 4.12 (s, 3 H, O-CH_3_), 3.81(m, 1 H, H-4′), 3.69 (m, 1 H, H-5′a), 3.67 (m, 1 H, H-5′b). ^13^C NMR (176 MHz): δ = 160.63 (C-6), 152.90 (C-4), 151.23 (C-2), 143.57 (C-8), 119.49 (C-5), 84.12 (C-4′), 84.10 (C-1′), 75.56 (C-2′), 74.23 (C-3′), 60.44 (C-5′), 54.87 (OCH_3_). ^15^N NMR (71 MHz): δ = 239.64 (N-7), 170.65 (N-9).

#### 2.8.7. 9-(β-D-Arabinofuranosyl)-2-fluoro-6-methoxypurine (**11**)

Thirty-six milligrams (0.12 mmol) of riboside (**6**) were dissolved in 10 mL of 10 mM KH_2_PO_4_ (pH 7.0). Fifty microliters of 20 mM Na_2_HAsO_4_ and *Ec*PNP (0.068 mg, 15 units, concentration in the reaction mixture 6.8 µg/mL) were added. The solution was heated up to 50 °C and kept for 3 h until the complete conversion of riboside (**6**) to base (**7**) (monitored by HPLC System 3, RT(**6**) = 7.9 min, RT(**7**) = 6.2 min). The reaction mixture was concentrated twice, cooled down to 4 °C, and kept for 16 h. The supernatant was removed, and the precipitate was washed with cold (4 °C) 10 mM KH_2_PO_4_ (pH 7.0) (2 × 5 mL) and suspended in 80 mL of 20 mM Tris∙HCl, pH 8.0. D-Arabinose (720 mg, 4.8 mmol), MnCl_2_ (25.2 mg, 0.2 mmol), KCl (298 mg, 4 mmol), and KH_2_PO_4_ (21.4 mg, 0.16 mmol) were added under stirring and heating at 50 °C. ATP (121 mg, 0.2 mmol), *Ec*RK (2.58 mg, 800 units, concentration in the reaction mixture 32 µg/mL), *Ec*PPM (32.7 mg, 160 units, concentration in the reaction mixture 408 µg/mL), and *Ec*PNP (6.55 mg, 1440 units, concentration in the reaction mixture 82 µg/mL) were added, and the reaction mixture was incubated at 50 °C for 48 h (monitored by HPLC System 3). The solution was neutralized with 2-N hydrochloric acid and concentrated *in vacuo* to ca. 35 mL. The precipitate was filtered off. The filtrate was applied to the column (octadecyl–Si 100 polyol (0.03 mm); 4 × 16 cm), and the product was eluted with a gradient of acetonitrile in water (0–50%, 1 L, flow rate 6 mL/min). Yield 15 mg (0.05 mmol, 42%), white amorphous powder, purity 99.0% (HPLC System 3, *R*_t_ = 7.7 min).

UV (H_2_O, pH 7.0) λmax, nm (ε, M^−1^cm^−1^): 254 (7200), 244 (6050).

HRMS (ESI+): *m*/*z* [M + H]^+^ calc. for C_11_H_14_N_4_O_5_F 301.0943; found 301.0927; [Base + H]^+^ calc. for C_6_H_6_N_4_OF 169.0520; found 169.0503.

^1^H NMR (700 MHz): δ = 8.46, (s, 1 H, H-8), 6.22 (d, *J* = 5.4, 1 H, H-1′), 5.72 (br. s., 1 H, OH-2′), 5.63 (br. s, 1 H, OH-3′), 5.11 (br. s, 1 H, OH-5′), 4.23 (t, *J* = 5.2, 1 H, H-2′), 4.14 (t, *J* = 5.4, 1 H, H-3′), 4.12 (s, 3 H, O-CH_3_), 3.80(m, 1 H, H-4′), 3.70 (dd, *J* = 3.7, 12.0, 1 H, H-5′a), 3.66 (dd, *J* = 4.8, 12.0, 1 H, H-5′b). ^13^C NMR (176 MHz): δ = 161.98 (d, *J*_C6,F_ = 17.0 Hz, C-6), 156.98 (d, *J*
_C2,F_ = 210.8 Hz, C-2), 153.00 (d, *J*_C4,F_ = 18.8 Hz, C-4), 143.38 (C-8), 118.64 (d, *J*_C5,F_ = 4.5 Hz, C-5), 84.09 (C-4′), 84.05 (C-1′), 75.59 (C-2′), 74.18 (C-3′), 60.44 (C-5′), 54.97 (OCH_3_). ^15^N NMR (71 MГц): δ = 239.9 (N-7), 200.11 (N-3), 170.71 (N-9).

## 3. Results

To create a polyenzymatic cascade for the synthesis of modified nucleosides, it was necessary to obtain a sufficient amount of recombinant ribokinase, phosphopentomutase, and nucleoside phosphorylases. We believe that the enzymes from *Escherichia coli* and thermophilic bacteria *Thermus* species 2.9 and *Thermus thermophilus* HB27 have the best prospects for research and subsequent use. The purifications of all the enzymes (except *Tth*PPM) were obtained earlier, the methods of their isolation were optimized, and the substrate characteristics were studied [21,22,23,24,28,29].

To obtain *Tth*PPM, an expression vector was constructed. The TT_RS08405 gene obtained by the amplification of genomic DNA from *Thermus thermophilu*s HB27 was cloned into the pET-23a+ vector. A strain *E. coli* NiCo21(DE3)/pER-PPM-Tth producing a soluble-form enzyme was created, and a purification protocol was developed. The technique includes the stages of heat precipitation of contaminating proteins and DNA, anion exchange, metal chelate affinity, and size-exclusion chromatography.

Preparations of purified enzymes were characterized by the purity, the content of oligomeric forms, and enzymatic activity (Table 1).

### 3.1. Study of the Influence of Various Factors on TthPPM Activity

The effect of pH, reaction mixture temperature, divalent cations, and cofactor (glucose-1,6- bisphosphate) on *Tth*PPM activity was studied. The enzymatic activity of *Tth*PPM was calculated, allowing for the content of impurity proteins.

The studied phosphopentomutase showed the maximal activity at pH 8.0 (Figure 1a). The enzyme was active in a wide temperature range; the highest activity was observed in the temperature range from 80 to 90 °C (Figure 1b).

The measurement of activity in the presence of chlorides of various metals showed that phosphopentomutase can bind various divalent cations, while the maximal activity was observed in the presence of a manganese cation. In the absence of divalent metal cations, the activity of the enzyme significantly decreased (Figure 2a). The studied phosphopentomutase exhibited maximal activity at a concentration of manganese ions of 0.5 mM (Figure 2b).

Glucose-1,6-bisphosphate increased the activity of *Tth*PPM (Figure 3), although its presence in the reaction was not necessary.

### 3.2. Determination of the Kinetic Parameters for TthPPM

The kinetic parameters were determined with natural substrates: D-ribose-5-phosphate (R5P) and 2-deoxy-D-ribose-5-phosphate (dR5P) (Table 2 and Figure 4). 

The data obtained indicate that the affinity of D-ribose-5-phosphate and 2-deoxy-D-ribose-5-phosphate for the active site of *Tth*PPM differs insignificantly. Thus, the presence of an OH group in the second position does not significantly affect the binding to the active site. At the same time, the reaction rate in the active site for D-ribose-5-phosphate is 7.1 times higher.

The activity of all the enzymes used in this study was determined (Table 3). For thermophilic enzymes, the activity was determined at 80 °C and, for mesophilic enzymes, at 37 °C. As the activity of *T*spRK was 16 times lower compared to *Ec*RK, more proteins were added to the reaction mixture. In this case, the activity of phosphopentomutase and purine nucleoside phosphorylase II (with respect to adenosine) of *Thermus thermophilus* was higher than that of *E. coli* enzymes, while the activity of *Tth*PNPI (with respect to inosine) and *Ec*PNP was the same.

### 3.3. Synthesis of Modified Heterocyclic Bases for Cascade Synthesis of Nucleosides

The use of a thermophilic cascade could be advantageous in the synthesis of 2,6-disubstituted purine arabinosides, since the solubility of 2-chloro- and 2-fluoradenine (for example) is known to be under 0.4 mM at 50 °C. Model reactions to determine the effectiveness of the thermophilic and mesophilic cascades were performed using 2-chloroadenine as a pentose acceptor.

After identifying the polyenzymatic cascade conditions, it was useful to synthesize both already known nucleosides (8-azaguanosine and allopurinol riboside) and new modified nucleosides using the cascade approach.

We decided to synthesize two bases: 2-chloro- and 2-fluoro-6-methoxypurine, and to obtain the corresponding arabinosides by a cascade of enzymes. From our experience, the optimal approach to the synthesis of 2-chloro- and 2-fluoro-6-methoxypurines is chemical–enzymatic (Figure 5 and Figure 6). This variant of the synthesis has not been previously described in the literature.

All chemical transformations of bases were carried out on ribonucleosides with acetyl protection of the ribose residue. In this case, the protected ribose acts as a protection for the highly reactive N9 in purine. 

In addition, the solubility of the nucleosides (both protected and unprotected) in most organic solvents is significantly higher than the solubility of the corresponding bases.

#### 3.3.1. Synthesis of 2-Chloro-6-methoxypurine (**3**)

The scheme of 2-chloro-6-methoxypurine (**3**) synthesis is shown in Figure 5. 

Riboside (**2**) was prepared from 9-(2,3,5-tri-O-acetyl-D-ribofuranosyl)-2,6-dichloropurine (**1**) by a treatment with sodium methylate in methanol, followed by crystallization from the reaction mixture (after neutralization with HCl). Additional purification was performed by column chromatography. Riboside (**2**) was obtained with 44% yield and 92.9% purity (HPLC). 

Base (**3**) was obtained from nucleoside (**2**) using an enzymatic arsenolysis of the glycosidic bond [32]. 

*E. coli* PNP can perform arsenolysis of a nucleoside bond. 1-α-Ribose arsenate is formed in the active site instead of 1-α-phosphate. Ribose arsenate is rapidly hydrolyzed to ribose and inorganic arsenate, and the equilibrium of the enzymatic reaction shifts towards the formation of base (**3**). *E. coli* PNP and Na_2_HAsO_4_ were added to the riboside (**2**) solution. The reaction mixture was incubated at 60 °C for several hours. Then, the mixture was concentrated to a minimal volume and kept at 4 °C for a while. The precipitate (2-chloro-6-methoxypurine (**3**)) was used in the synthesis of arabinoside without additional purification.

#### 3.3.2. Synthesis of 2-Fluoro-6-methoxypurine (**7**)

The starting compound for the preparation of 2-fluoro-6-methoxypurine (**7**) was 2-amino-6-chloropurine riboside triacetate (**4**), obtained by the method [27]. The substitution of the 2-amino group with fluorine was performed by the action of 70% HF in pyridine (Figure 6).

Product (**5**) was isolated by column chromatography on silica gel. 2-Fluoro-6-methoxyriboside (**6**) was prepared by a treatment with sodium methylate in methanol according to the procedure described above. The target riboside of 2-fluoro-6-methoxypurine was isolated by column chromatography on a silica gel. The yield of nucleoside (**6**) was 44%, and the byproduct, according to NMR data, was 2,6-dimethoxypurine riboside. The target product (**7**) was synthesized by the same method as base (**3**) and used in the subsequent transformation without purification.

### 3.4. Conditions for Poly-Enzymatic Cascades Using Mesophilic and Thermophilic Enzymes

A series of poly-enzymatic cascades using recombinant thermophilic enzymes and *E. coli* enzymes was performed. The cascade of enzymatic reactions includes the sequential conversion of D-pentoses to 5-monophosphates, catalyzed by ribokinase (RK), the conversion of 5-phosphates to α-D-pentose-1-phosphates, catalyzed by phosphopentomutase (PPM), and the condensation of them with the corresponding base, catalyzed by purine nucleoside phosphorylase (PNP). This leads to the formation of the desired β-D-nucleoside (Figure 7). The reactions were monitored using the liquid chromatography–mass spectrometry (LC-MS) method.

The conditions for the polyenzymatic cascade with mesophilic *E. coli* enzymes were determined earlier [18,33]: temperature 50 °C, pH 8.0, and the presence of potassium chloride and manganese ions. In the case of thermophilic enzymes, the conditions differed only in temperature (80 °C).

The conditions for performing the poly-enzymatic cascade with recombinant thermophilic enzymes with various D-pentoses are presented in Table 4.

When selecting conditions for cascade reactions, previous experience with these enzymes was taken into account [18,22,33]. High concentrations of arabinose and xylose were due to their poor affinity for the active site of ribokinase. In addition, at ATP concentrations above 3 mM, ribokinase inhibition is observed.

As we expected, at first glance, the conditions for conducting polyenzymatic cascades of nucleoside synthesis are very similar, except for the temperature of reaction mixtures. However, there is a significant difference in synthesis efficiency and the stability of products under the conditions of obtaining 2-chloroadenine nucleosides.

### 3.5. Synthesis of Nucleosides *(**8**–**11**)*

We attempted to synthesize ribosides of two bases: 8-azaguanine (8-Aza-Gua) and allopurinol (Allop), using both mesophilic and thermophilic pathways. The conditions for the process are presented in Table 5. 

For 8-azaguanine, the efficiency of the thermophilic cascade differed slightly from the mesophilic one. Therefore, we decided to synthesize 8-azaguanosine (**8**) using thermophilic enzymes, since the increased temperature provided acceptable solubility of the starting heterocycle. *E. coli* enzymes were used to obtain the allopurinol riboside (**9**), as they provide higher conversion of the base to riboside (96% per day).

The products were isolated from the reaction mixtures by column chromatography. The yields of nucleosides (**8**) and (**9**) were 69% and 84%, respectively.

The main conditions for the cascade synthesis of nucleosides (**8**–**10**) are shown in Table 6.

The synthesis of 9-(β-D-Arabinofuranosyl)-2-chloro-6-methoxypurine (**10**) and 9-(β-D-Arabinofuranosyl)-2-fluoro-6-methoxy-purine (**11**) was performed without preliminary optimization of the conditions. Only the quantity of *E. coli* enzymes was doubled. This allowed achieving the conversion of the base into arabinoside for the 2-chloro derivative (**10**) of 65% and for the fluorinated analog (**11**) of 90% in two days.

The products were isolated by column chromatography. The yields of nucleosides (**10**) and (**11**) were 54% and 42%, respectively.

## 4. Discussion

To understand the peculiarities of the mesophilic and thermophilic cascades in the synthesis of 2-chloroadenine riboside, 2-deoxyriboside, arabinoside, 2-deoxy-2-fluoro-D-arabinoside, and xyloside, it is necessary to scrutinize the dynamics of nucleoside formation.

### 4.1. Possibilities of Using Various D-Pentoses in the Synthesis of 2-Chloroadenine Nucleosides

D-ribose, 2-deoxy-D-ribose, and their phosphates are natural substrates of nucleic acid and carbohydrate metabolism enzymes. Therefore, their conversion in a cascade with thermophilic enzymes proceeded quite intensively. The maximal content of 2-chloroadenosine (Ribo-2ClAde) was 78% after 30 min, and 2′-deoxy-2-chloroadenosine (dRibo-2ClAde) was 60% after 30 min (Figure 8). Interestingly, the equilibrium of the cascade synthesis in the synthesis of the 2-chloroadenine riboside by *E. coli* enzymes was shifted towards nucleoside formation. However, under the conditions of a thermophilic cascade, the maximal product concentration was observed after 30 min, and then, the equilibrium began to shift in the opposite direction, which led to a twofold decrease of product content in the reaction mixture. These results may be due to the increasing hydrolysis rate of α-D-ribose-1-phosphate with the temperature increasing [34].

The content of 2′-deoxy-2-chloroadenosine also sharply decreased over time due to the equilibrium shift towards the opposite reaction because of the hydrolysis of 2-deoxy-α-D-ribose-1-phosphate (Figure 8b).

In the cascade reaction with *E. coli* enzymes, the maximal content of 2-chloroadenosine (91%) was observed after one hour and 2′-deoxy-2-chloroadenosine (73%) after 30 min. The content of 2-chloroadenosine in the reaction mixture remained constant. The content of 2′-deoxy-2-chloroadenosine gradually decreased over time due to the shift of equilibrium towards the opposite reaction. However, the rate of the opposite reaction was noticeably lower than in the cascade with thermophilic enzymes.

The conversion of D-xylose in the thermophilic cascade was very slow, and the maximal conversion of 2-chloroadenine to 9-(β-D-xylofuranosyl)-2-chloroadenine was only 3.9% after 24 h (Figure 9a). The maximal content of 9-(β-D-xylofuranosyl)-2-chloroadenine was 26% after 24 h in the mesophilic cascade, which is much higher than in the cascade with thermophilic enzymes.

The formation of 9-(β-D-arabinofuranosyl)-2-chloroadenine from D-arabinose proceeded much more slowly (Figure 9a) compared to the formation of 2-chloroadenosine and 2′-deoxy-2-chloroadenosine (arabinoside content 48.1% after 24 h). The maximal content of 9-(β-D-arabinofuranosyl)-2-chloroadenine was higher in the mesophilic cascade than in the cascade with thermophilic enzymes (67% after an hour), but then, its content gradually decreased.

The conversion of 2-deoxy-2-fluoro-D-arabinose to 9-(2′-deoxy-2′-fluoro-β-D-Arabinofuranosyl)-2-chloroadenine (clofarabine) (Figure 9b) proceeded slower than the synthesis of arabinosides (the maximal content was 19% after 24 h, from a thermophilic cascade). The conversion of 2-deoxy-2-fluoro-D-arabinose to clofarabine proceeded much better in the mesophilic cascade than in the cascade with thermophilic enzymes (the maximal content of clofarabine was 46% versus 19% after 24 h).

Kamel with coworkers [35] investigated the stability of some pentose-1-phosphates at various temperatures. The stability of α-D-arabinose-1-phosphate was higher than that of α-D-ribose-1-phosphate and 2-deoxy-α-D-ribose-1-phosphate, but all these compounds were hydrolyzed. The stability of 2-deoxy-2-fluoro-α-D-arabinose-1-phosphate was much higher than that of all other pentose-1-phosphates. A slow decreasing of the product content after 24h in the cascade with D-arabinose and the absence of decreasing in the cascade with 2-deoxy-2-fluoro-D-arabinose may be due to a low rate of 1-phosphate intermediate hydrolysis.

The lower rate of D-arabinosides synthesis in comparison with D-ribosides and 2-deoxy-D-ribosides can be explained by the different conformations of carbohydrates in the active site of purine nucleoside phosphorylase. For this reason, the rate of D-arabinosides phosphorolysis is also much lower [36].

### 4.2. Non-Natural Heterocyclic Bases in the Cascade Synthesis of Nucleosides

To determine the possibility of practical use of thermophilic or mesophilic cascades, we tried to synthesize several non-natural nucleosides. We selected 8-azaguanine, allopurinol, 2-chloro-6-methoxypurine (**3**), and 2-fluoro-6-methoxypurine (**7**) as the heterocyclic bases, taking into account the prospect of subsequent biological testing of the corresponding modified nucleosides.

#### Synthesis of 8-Azaguanine and Allopurinol Ribosides

We investigated the formation of 9-(β-D-ribofuranosyl)-8-azaguanine and 9-(β-D-ribofuranosyl) allopurinol from 8-azaguanine and allopurinol, respectively. In both cases, the transformation proceeded at a high rate. The formation of 9-(β-D-ribofuranosyl) allopurinol (Figure 10a) in the reaction with *E. coli* enzymes proceeded at a high rate; its maximal content was 95% after eight hours and did not change over time. The maximal content of 9-(β-D-ribofuranosyl) allopurinol in the thermophilic cascade was 73% after eight hours; then, the content decreased. The rate of conversion of 8-azaguanine to 8-azaguanosine (8-azaGuo) in the mesophilic cascade was higher than in the cascade with thermophilic enzymes (Figure 10b). 8-Azaguanosine gradually accumulated in the reaction mixture with *E. coli* enzymes, and its maximum content was 89% after 24 h.

Using the parameters of the test cascade syntheses, four nucleosides were prepared and isolated. 8-Azaguanosine was synthesized by a thermophilic cascade with a 69% yield. Allopurinol riboside was obtained by a mesophilic cascade with an 84% yield. For the first time, 2-chloro-6-methoxypurine (**3**) and 2-fluoro-6-methoxypurine (**7**) D-arabinosides were synthesized using a mesophilic cascade (yields 54% and 41%, respectively).

All synthesized nucleosides were characterized by LC-MS data. The structure was confirmed by NMR spectra: ^1^H, ^1^H-^1^H-COSY, ^1^H-^13^C-HSQC, ^1^H-^13^C-HMBC, ^1^H-^15^N-HSQC, and ^1^H-^15^N-HMBC.

The data obtained indicated that, despite the relatively small difference in temperatures when performing the cascade reactions (50 and 80 °C), the rate of product formation in the reactions with *E. coli* enzymes was much higher. The *E. coli* enzymes also provided a higher content of target products in the reaction mixture. Therefore, they were more appropriate for use in the polyenzymatic synthesis of modified nucleosides.

## 5. Conclusions

To carry out a comparative study of the possibilities of using the RK → PPM → NP cascade in the synthesis of modified nucleosides, sufficient quantities of genetically engineered enzymes of both *E. coli* and thermophilic *Thermus* species 2.9 and *Thermus thermophilus* HB27 were obtained. Recombinant phosphopenthomutase PPM from *Thermus thermophilus* HB27 was obtained for the first time: a strain-producer of the soluble form of the enzyme was created, and a procedure for its isolation and chromatographic purification was developed. The influence of the pH, temperature, presence of divalent cations, and cofactor (glucose-1,6-bisphosphate) on the activity of *Tth*PPM was studied. The maximum activity was observed at pH 8.0. The enzyme is active over a wide temperature range, with the maximum activity in the interval between 80 and 90 °C. Glucose-1,6-bisphosphate significantly increased the activity of *Tth*PPM, although its presence in the reaction was not necessary.

The preparations of all purified enzymes were characterized by purity, the content of the oligomeric forms, and enzymatic activity.

It was shown that the cascade synthesis of the modified nucleosides from D-pentoses could be performed by both the mesophilic and thermophilic enzymes. The following D-pentoses were tested: ribose, 2-deoxyribose, arabinose, xylose, and 2-deoxy-2-fluoroarabinose. The efficiency of 2-chloroadenine nucleoside synthesis with mesophilic enzymes decreased in the following order: Rib (92), dRib (74), Ara (66), F-Ara (8), and Xyl (2%) in 30 min. For thermophilic enzymes, the order was Rib (76), dRib (62), Ara (32), F-Ara (<1), and Xyl (2%) in 30 min. After one day of incubation, the amounts of 2-chloroadenine riboside (thermophilic cascade), 2-deoxyribosides (both cascades), and arabinoside (mesophilic cascade) decreased roughly by half. Meanwhile, the conversion of the base to 2-fluoroarabinosides and xylosides continued to grow in both cases and reached 20-40%. That was quite acceptable for the preparative synthesis of nucleosides.

Four D-ribose and D-arabinose nucleosides were synthesized on a large scale by a cascade of enzymes. 8-Azaguanine riboside was synthesized by the thermophilic cascade with a 69% yield. Allopurinol riboside was obtained by the mesophilic cascade with an 84% yield. For the first time, 2-chloro-6-methoxypurine and 2-fluoro-6-methoxypurine D-arabinosides were synthesized using the mesophilic cascade (yields 54% and 42%, respectively). The data obtained indicated that, despite the relatively small difference in temperatures when performing the cascade reactions (50 and 80 °C), the rate of the product formation in the reactions with *E. coli* enzymes was much higher. The *E. coli* enzymes also provided a higher content of products in the reaction mixtures. Therefore, they are more appropriate for use in the polyenzymatic synthesis of modified nucleosides. The use of thermophilic enzymes and a high reaction temperature might be preferable with low-soluble heterocyclic bases (2-chloroadenine and 2-fluoroadenine), as well as in the case of using carbohydrates that form 1-phosphates with high temperature stability (D-arabinose and 2-deoxy-2-fluoro-D-arabinose).

## Data Availability

The data presented in this study are available on request from the corresponding author.

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
