# Peer review of "Multi-Enzymatic Cascades in the Synthesis of Modified Nucleosides: Comparison of the Thermophilic and Mesophilic Pathways"

_biomolecules, 2021, doi:10.3390/biom11040586_

Round 1

Reviewer 1 Report

The manuscript “Multyenzymatic cascades in the synthesis of modified nucleosides: comparison of the thermophilic and mesophilic pathways” by Fateev and coworkers describes an interesting study where the biosynthetic utility of series of nucleoside modifying enzymes was investigated.

Overall, the results obtained are very interesting.

However, in the introduction, the authors refer to biosynthetically derived pyrimidine nucleosides and their biosynthesis, without citing sufficient literature. Also they refer to chlorinated and fluorinated nucleosides, but certain very important recent reports on the topic are not included. The following papers on the synthesis and biosynthesis of nucleosides, including fluorinated analogues and enzymatic processes, should be cited:

  1. Natural Products Incorporating Pyrimidine Nucleosides, https://doi.org/10.1016/B978-0-12-409547-2.14797-3
  2. Pac13 is a Small, Monomeric Dehydratase that Mediates the Formation of the 3'-Deoxy Nucleoside of Pacidamycins, doi: 10.1002/anie.201705639
  3. Synthesis and Conformational Analysis of Fluorinated Uridine Analogues Provide Insight into a Neighbouring-Group Participation Mechanism, https://doi.org/10.3390/molecules25235513

Regarding figure 9a and lines 601-602 more clarification is needed: the authors mention that “The content of 2ʹ‐deoxy‐2‐chloroadenosine gradually decreased over time due to shift of equilibrium towards the opposite reaction.” Could the authors please clarify/further support this claim? Could they rule out product decomposition under the reported reaction conditions?

Author Response

Reviewer 1.

However, in the introduction, the authors refer to biosynthetically derived pyrimidine nucleosides and their biosynthesis, without citing sufficient literature. Also they refer to chlorinated and fluorinated nucleosides, but certain very important recent reports on the topic are not included. The following papers on the synthesis and biosynthesis of nucleosides, including fluorinated analogues and enzymatic processes, should be cited:

  1. Natural Products Incorporating Pyrimidine Nucleosides, https://doi.org/10.1016/B978-0-12-409547-2.14797-3
  2. Pac13 is a Small, Monomeric Dehydratase that Mediates the Formation of the 3'-Deoxy Nucleoside of Pacidamycins, doi: 10.1002/anie.201705639
  3. Synthesis and Conformational Analysis of Fluorinated Uridine Analogues Provide Insight into a Neighbouring-Group Participation Mechanism, https://doi.org/10.3390/molecules25235513
  • Citations was added to introduction: [7] for first paper, [8] for second paper, [17] for third paper.

Regarding figure 9a and lines 601-602 more clarification is needed: the authors mention that “The content of 2`‐deoxy‐2‐chloroadenosine gradually decreased over time due to shift of equilibrium towards the opposite reaction.” Could the authors please clarify/further support this claim? Could they rule out product decomposition under the reported reaction conditions?

  • Rate of hydrolysis of α-D-ribose-1-phosphate and 2-deoxy-α-D-ribose-1-phosphate depends on temperature (the higher temperature, the higher rate). Content of the target nucleoside is decreased due to decomposition of α-D-ribose-1-phosphate and 2-deoxy-α-D-ribose-1-phosphate. Hydrolysis kinetics of ribose-1-phosphate was reported by Bunton (https://doi.org/1021/jo01255a019).
  • Citation of Bunton work was added to paragraph 3.5: “These results may be due to increasing hydrolysis rate of α-D-ribose-1-phosphate with temperature increasing [35]”.

Reviewer 2 Report

The authors describe in the manuscript entitled “Multyenzymatic cascades in the synthesis of modified nucleosides: comparison of the thermophilic and mesophilic pathways” the cloning, expression, and characterization of the thermostable PPM of Thermus thermophilus. Furthermore, the authors compare enzymatic cascade reactions consisting of either mesophilic or thermophilic enzymes. In my opinion this is an interesting approach, however, the authors have not addressed important points in their work: The main point is, why authors compare different NPs for the synthesis for a wide variety of nucleoside analogues? Nucleoside phosphorolysis is a thermodynamically controlled reaction and the final yield of such a transformation depends solely on the nucleoside in question and not on the enzymes used. Furthermore, in this study it was shown that product yield decreased for some compounds because of the hydrolysis of the pentose-1-phosphate intermediates. It is well known that temperature has a strong impact on pentose-1-phosphate stability, hence, authors should have used the same temperatures for the mesophilic and thermophilic cascade when handling instable intermediates.

In summary, the manuscript is unfortunately very difficult to read. It would certainly simplify the reading if comparative illustrations were created that summarize the results of one topic. In addition, it would be easier to interpret the results if experimental data were also included to the figures/tables. Unfortunately, the discussion suffers from the fact that the results are evaluated rather than placed in the scientific context.

Therefore, I recommend a major revision including a significant language revision.

Specific comments:

Please correct the spelling error in the title: “Multyenzymatic” in "Multienzymatic”.

In line 61, the authors state that most NPs retain 70% of their activity at 60 °C. This is an over-generalized statement as all NPs are subject to denaturation at this temperature. However, the denaturation rate at 60°C varies greatly, lending some NPs tremendous stability.

The authors describe the use of Tris-HCl buffer, which may have been an unfortunate choice for reactions at 80°C since Tris is known to have a very temperature-dependent pKa and buffer capacity. I would assume that the reaction mixtures at 80°C were in fact much more acidic than pH 8.

I would recommend the addition of details regarding enzyme concentration in the experimental procedures.

Figure 1b shows clear evidence of enzyme denaturing impacting the activity measurements. Can the authors provide a melting point for the enzymes in question?

Considering the experimental conditions used in the reactions for Figure 1, it appears likely that the glycosylation reaction used as an indicator did reach close to equilibrium, which creates a disproportionality between “real” activity and measured conversion. Could the authors provide additional details for this procedure and ideally raw data?

The horizontal axis in Figure 2a says pH but displays various metal ions.

Please provide direct Michaelis-Menten fits for the data in Figure 4, since Lineweaver-Burk plots can suffer from errors due to linearization of the data. The data should also be transferred to the SI.

The paragraph in lines 460-465 needs to include the differences in reaction temperature, since this is a tremendously important variable for activity.

The syntheses described in Figure 5 and 6 should be reconsidered. Although they do indeed describe “new” syntheses, they constitute no improvement over existing procedures, are significantly more laborious, have a much lower atom economy and use highly hazardous chemicals. Both target nucleosides have previously been prepared from the corresponding nucleobases in one-step procedures (e.g. 10.1021/acsmedchemlett.5b00064 and 10.1055/s-0036-1588134).

The data presented in tables 4-6 does not present an “optimization” of the enzymatic cascade. Please revise the description of these results.

Figure 8 and following show clear evidence for hydrolysis of deoxyribose- and ribose-1-phosphate at 80°C. Considering the lower pH in hot Tris buffer, the data shown in these figures roughly agree with the hydrolysis kinetics of ribose-1-phosphate reported by Bunton (10.1021/jo01255a019).

The data in Figure 9 and following also show that the reactions with the thermostable enzymes had not reached completion, which casts doubt on the primary conclusion of the article that the mesophilic enzymes used are “more appropriate”.

To me, it is unclear why the authors chose this particular thermostable PNP variants as previous studies on these enzymes had revealed a rather narrow substrate spectrum.

Author Response

Reviewer 2

The authors describe in the manuscript entitled “Multyenzymatic cascades in the synthesis of modified nucleosides: comparison of the thermophilic and mesophilic pathways” the cloning, expression, and characterization of the thermostable PPM of Thermus thermophilus. Furthermore, the authors compare enzymatic cascade reactions consisting of either mesophilic or thermophilic enzymes. In my opinion this is an interesting approach, however, the authors have not addressed important points in their work: The main point is, why authors compare different NPs for the synthesis for a wide variety of nucleoside analogues?

  • Added to the end of introduction:

“The synthesis of natural nucleosides using various nucleoside phosphorylases gives similar results. However, for modified heterocyclic bases or different carbohydrates, the results can vary significantly.”

Nucleoside phosphorolysis is a thermodynamically controlled reaction and the final yield of such a transformation depends solely on the nucleoside in question and not on the enzymes used. Furthermore, in this study it was shown that product yield decreased for some compounds because of the hydrolysis of the pentose-1-phosphate intermediates. It is well known that temperature has a strong impact on pentose-1-phosphate stability, hence, authors should have used the same temperatures for the mesophilic and thermophilic cascade when handling instable intermediates.

  • Added to the end of introduction:

“In addition, some heterocyclic bases, such as 2-chloroadenine and 2-fluoroadenine, have low water solubility and the use of thermophilic enzymes may be preferable. Carrying out the reaction with thermophilic enzymes at the operating temperatures of mesophilic enzymes reduces the usefulness of this approach; therefore, the reactions with Thermus thermophilus enzymes were carried out at a higher temperature.”

In summary, the manuscript is unfortunately very difficult to read. It would certainly simplify the reading if comparative illustrations were created that summarize the results of one topic. In addition, it would be easier to interpret the results if experimental data were also included to the figures/tables.

  • Experimental data added to table 2.

Unfortunately, the discussion suffers from the fact that the results are evaluated rather than placed in the scientific context.

Therefore, I recommend a major revision including a significant language revision.

Specific comments:

Please correct the spelling error in the title: “Multyenzymatic” in "Multienzymatic”.

  • Title was corrected.

In line 61, the authors state that most NPs retain 70% of their activity at 60 °C. This is an over-generalized statement as all NPs are subject to denaturation at this temperature. However, the denaturation rate at 60°C varies greatly, lending some NPs tremendous stability.

  • “most NPs retain more than 70% of their activity” changed to “some NPs retain most of their activity”.

The authors describe the use of Tris-HCl buffer, which may have been an unfortunate choice for reactions at 80°C since Tris is known to have a very temperature-dependent pKa and buffer capacity. I would assume that the reaction mixtures at 80°C were in fact much more acidic than pH 8.

  • You are right. Our experimental data on the pH of 20 mM Tris·HCl versus temperature is presented in the graph below. At 25°C pH 8.0, at 50°C pH 7.3 and at 80°C pH 6.4. The optimal conditions for the enzymatic reactions were determined in this buffer (for example, Fig. 1a). We will consider the possibility of replacing the buffer in future work.

I would recommend the addition of details regarding enzyme concentration in the experimental procedures.

  • Enzyme concentrations added to the experimental procedures.

Figure 1b shows clear evidence of enzyme denaturing impacting the activity measurements. Can the authors provide a melting point for the enzymes in question?

  • We have not performed experiments to determine the melting temperature.
  • We have no kinetics data (kcat for various temperatures) for calculation of KRD(T) and Tm as described in Hei and Clark paper (https://doi.org/10.1002/bit.260421015).

Considering the experimental conditions used in the reactions for Figure 1, it appears likely that the glycosylation reaction used as an indicator did reach close to equilibrium, which creates a disproportionality between “real” activity and measured conversion. Could the authors provide additional details for this procedure and ideally raw data?

  • Mass of enzymes in reaction mixtures (Figure 1): 0.1 μg TthPPM and 2.7 μg TthPNPII (0.0014 units TthPPM and 0.54 units TthPNPII) in each reaction mixture (unit = µmol/min). Thus, the reaction with TthPPM is the rate-limiting stage. Reaction time is 2 minutes. The amount of enzymes was chosen for initial velocity measurement (after two minutes the reaction was far from equilibrium). For example, at 80°C 2.2% of adenine was converted to adenosine.

The horizontal axis in Figure 2a says pH but displays various metal ions.

  •  

Please provide direct Michaelis-Menten fits for the data in Figure 4, since Lineweaver-Burk plots can suffer from errors due to linearization of the data. The data should also be transferred to the SI.

  • Michaelis-Menten plots were added to Figure 4. Data was transferred to the SI both for Michaelis-Menten and Lineweaver-Burk plots.

The paragraph in lines 460-465 needs to include the differences in reaction temperature, since this is a tremendously important variable for activity.

  • Text was added: “For thermophilic enzymes, the activity was determined at 80 °C, and for mesophilic enzymes at 37 °C”.

The syntheses described in Figure 5 and 6 should be reconsidered. Although they do indeed describe “new” syntheses, they constitute no improvement over existing procedures, are significantly more laborious, have a much lower atom economy and use highly hazardous chemicals. Both target nucleosides have previously been prepared from the corresponding nucleobases in one-step procedures (e.g. 10.1021/acsmedchemlett.5b00064 and 10.1055/s-0036-1588134).

  • Paragraph 3.3: “We decided to synthesize two new bases” – “new” was removed.
  • “This variant of synthesis has not been previously described in the literature.” We meant that these heterocyclic bases were not previously obtained using arsenolysis. And the chemical synthesis of nucleosides is not new. Thank you for papers.

The data presented in tables 4-6 does not present an “optimization” of the enzymatic cascade. Please revise the description of these results.

  • Paragraph 3.3: “After identifying the most effective polyenzymatic cascade it was interesting to…” was changed to “After identifying the polyenzymatic cascade conditions it was interesting to…”.
  • Paragraph 3.4:
  • “3.4. Determination of optimal conditions for poly-enzymatic…” was changed to “3.4. Conditions for poly-enzymatic…”
  • Before and after Table 4 – “optimal” was removed.
  • Table 4 caption – “optimal” was removed.
  • Paragraph 3.5: before Table 5 – “optimal” was removed.
  • Paragraph 4.2: “to optimize and” was removed.

Figure 8 and following show clear evidence for hydrolysis of deoxyribose- and ribose-1-phosphate at 80°C. Considering the lower pH in hot Tris buffer, the data shown in these figures roughly agree with the hydrolysis kinetics of ribose-1-phosphate reported by Bunton (10.1021/jo01255a019).

  • Citation of Bunton work was added to paragraph 3.5: “These results may be due to increasing hydrolysis rate of α-D-ribose-1-phosphate with temperature increasing [35]”.

The data in Figure 9 and following also show that the reactions with the thermostable enzymes had not reached completion, which casts doubt on the primary conclusion of the article that the mesophilic enzymes used are “more appropriate”.

  • In Figure 9b reaction with the thermostable enzymes did not reach equilibrium, but % of nucleoside is more than half less, than for coli enzymes and slow growth is observed. After some days, no ATP remains in the reaction mixture and hydrolysis of 2-fluoro- α-D-arabinose-1-phosphate does not allow to get a large amount of product. In Figure 10b, the synthesis efficiency is approximately at the same level.

To me, it is unclear why the authors chose this particular thermostable PNP variants as previous studies on these enzymes had revealed a rather narrow substrate spectrum.

  • Various pentose-1-phosphates are substrates for these enzymes. That makes them potentially useful. All tested heterocyclic bases without bulky groups are also substrates. To expand the substrate specificity, it is possible to obtain mutant forms.

Round 2

Reviewer 2 Report

Thank you for responding in such detail to the questions of the first revision. I still consider the first part of the manuscript to be very good and it has also gained in value due to the additions.

However, I still disagree with the presentation of the discussion and the final conclusion of the paper. A single refrence has now been added to the discussion. In my eyes, this continues to not place the results in a scientific context. Specific comments on the discussion follow at the end.

Based on the results presented, I find it very difficult to conclude that mesophilic enzymes are better than thermophilic enzymes for the synthesis of nucleoside analogs. As mentioned in the first revision, the authors chose thermostable PNPs that show a relatively high substrate specificity. This may be accompanied by kinetic parameters that are significantly worse than for the mesophilic enzymes used. This is already evident from the presented specific activities of the PNPs. For inosines, a higher specific activity was observed with the thermostable enzyme while for adenosine the mesophilic enzyme showed a higher specific activity. In my opinion, kinetic parameters should also have been determined for the sugar-modified variants in order to find suitable conditions that lead to the same product formation rates.

While I understand the argument for using thermostable enzymes at higher temperatures to increase solubilities, it is not understandable for me to use higher reaction temperatures when instable intermediates are formed. In this case, the reaction conditions have to be adjusted in my opinion.

Therefore, a more balanced conclusion is in my opinion is a basic requirement for the acceptance of the presented manuscript.

Specific comments to the discussion:

The authors state: “The conversion of D-xylose in the thermophilic cascade was very slow, and the maximal conversion of 2-chloroadenine to 9-(β-D-xylofuranosyl)-2-chloroadenine was only 3.9% after 24 hours (Figure 9a). The maximal content of 9-(β-D-xylofuranosyl)-2-chloroadenine was 26% after 24 hours in the mesophilic cascade. That is much higher than in the cascade with thermophilic enzymes.”

This is just an observation, but what is the explanation? Is xylulose-1P accepted by the thermophilic PNP? If so, what are the kinetic data for the mesophilic and the thermostable PNP with xylulose-1P?

The authors state: “The maximal content of 9-(β-D-arabinofuranosyl)-2-chloroadenine was higher in the mesophilic cascade than in the cascade with thermophilic enzymes (67% after an hour), but then its content gradually decreased.”

Again the question, what is the explanation. Please present data from the literature where the stability of arabinose-1P was studied with increasing temperature.

Line 618-660. The authors just re-describe their results concerning the question which substrates are converted better than the others. A comparison to literature data is almost completely missing. Explanations should be presented why yields for ribose nucleosides is higher than for arabinose nucleosides (as an example). Data are available in the literature.

If comparing mesophilic and thermophilic cascades, which enzyme is the bottleneck for the synthesis? If the PNP is the bottleneck kinetic data should be presented to compare the mesophilic and thermophilic PNP with different sugar-modified adenosine derivatives. I would assume for synthesis routes with stable sugar-1-phosphate intermediates that kinetic parameters are worse for the tested thermostable enzymes compared to the mesophilic enzymes and, hence, equilibrium was not reached within in the given reaction time.

Author Response

Open Review

English language and style

( ) Extensive editing of English language and style required
(x) Moderate English changes required
( ) English language and style are fine/minor spell check required
( ) I don't feel qualified to judge about the English language and style

Comments and Suggestions for Authors

Thank you for responding in such detail to the questions of the first revision. I still consider the first part of the manuscript to be very good and it has also gained in value due to the additions.

However, I still disagree with the presentation of the discussion and the final conclusion of the paper. A single refrence has now been added to the discussion. In my eyes, this continues to not place the results in a scientific context. Specific comments on the discussion follow at the end.

Based on the results presented, I find it very difficult to conclude that mesophilic enzymes are better than thermophilic enzymes for the synthesis of nucleoside analogs. As mentioned in the first revision, the authors chose thermostable PNPs that show a relatively high substrate specificity. This may be accompanied by kinetic parameters that are significantly worse than for the mesophilic enzymes used. This is already evident from the presented specific activities of the PNPs. For inosines, a higher specific activity was observed with the thermostable enzyme while for adenosine the mesophilic enzyme showed a higher specific activity. In my opinion, kinetic parameters should also have been determined for the sugar-modified variants in order to find suitable conditions that lead to the same product formation rates.

While I understand the argument for using thermostable enzymes at higher temperatures to increase solubilities, it is not understandable for me to use higher reaction temperatures when instable intermediates are formed. In this case, the reaction conditions have to be adjusted in my opinion.

Therefore, a more balanced conclusion is in my opinion is a basic requirement for the acceptance of the presented manuscript.

  • English was corrected.
  • Text was added to the end of conclusions: “The use of thermophilic enzymes and a high reaction temperature might be preferable with low soluble heterocyclic bases (2-chloroadenine, 2-fluoroadenine), as well as in the case of using carbohydrates that form thermally stable 1-phosphates with high temperature stability (D-arabinose, 2-deoxy-2-fluoro-D-arabinose)”.

Specific comments to the discussion:

The authors state: “The conversion of D-xylose in the thermophilic cascade was very slow, and the maximal conversion of 2-chloroadenine to 9-(β-D-xylofuranosyl)-2-chloroadenine was only 3.9% after 24 hours (Figure 9a). The maximal content of 9-(β-D-xylofuranosyl)-2-chloroadenine was 26% after 24 hours in the mesophilic cascade. That is much higher than in the cascade with thermophilic enzymes.”

This is just an observation, but what is the explanation? Is xylulose-1P accepted by the thermophilic PNP? If so, what are the kinetic data for the mesophilic and the thermostable PNP with xylulose-1P?

  • We do not have xylose-1-phosphate to compare the activity of thermophilic and mesophilic PNP. There are no published data on the activity of nucleoside phosphorylases with xylose-1-phosphate or xylo-nucleosides.

The authors state: “The maximal content of 9-(β-D-arabinofuranosyl)-2-chloroadenine was higher in the mesophilic cascade than in the cascade with thermophilic enzymes (67% after an hour), but then its content gradually decreased.”

Again the question, what is the explanation. Please present data from the literature where the stability of arabinose-1P was studied with increasing temperature.

  • Reference to Kamel et al work was added. (https://doi.org/10.1016/j.mcat.2018.07.028).
  • Text was added to paragraph 4.1: “Kamel with co-workers [36] has investigated the stability of some pentose-1-phosphates at various temperatures. The stability of α-D-arabinose-1-phosphate was higher than that of α-D-ribose-1-phosphate and 2-deoxy-α-D-ribose-1-phosphate, but all these compounds were hydrolyzed. The stability of 2-deoxy-2-fluoro-α-D-arabinose-1-phosphate was much higher than that of all other pentose-1-phosphates. Slow decreasing of the product content after 24h in the cascade with D-arabinose and the absence of decreasing in the cascade with 2-deoxy-2-fluoro-D-arabinose may be due to low rate of 1-phosphate intermediate hydrolysis”.

Line 618-660. The authors just re-describe their results concerning the question which substrates are converted better than the others. A comparison to literature data is almost completely missing. Explanations should be presented why yields for ribose nucleosides is higher than for arabinose nucleosides (as an example). Data are available in the literature.

  • Reference to Bennett et al work was added. (https://doi.org/10.1074/jbc.M304622200).
  • Text was added to paragraph 4.1: “The lower rate of D-arabinosides synthesis in comparison with D-ribosides and 2-deoxy-D-ribosides can be explained by the different conformation of carbohydrates in the active site of purine nucleoside phosphorylase. For this reason, the rate of D-arabinosides phosphorolysis is also much lower [37]”.

If comparing mesophilic and thermophilic cascades, which enzyme is the bottleneck for the synthesis? If the PNP is the bottleneck kinetic data should be presented to compare the mesophilic and thermophilic PNP with different sugar-modified adenosine derivatives. I would assume for synthesis routes with stable sugar-1-phosphate intermediates that kinetic parameters are worse for the tested thermostable enzymes compared to the mesophilic enzymes and, hence, equilibrium was not reached within in the given reaction time.

  • In this work, we used five pentoses and six enzymes. One enzyme can be the bottleneck when using one carbohydrate, but not the bottleneck when using another carbohydrate. We do not have all the intermediates for determining the enzymatic activity in all cases.